# Salvage Post-Operative Stereotatic Ablative Radiotherapy for Re-Current Squamous Cell Carcinoma of Head and Neck

**DOI:** 10.3390/medicina58081074

**Published:** 2022-08-10

**Authors:** Antonio Pellizzon, Maria Silva, Ricardo Fogaroli, Elson Neto, Michael Chen, Guilherme Godim, Douglas Castro, Henderson Ramos, Carolina Abrahao

**Affiliations:** AC Camargo Cancer Center, Sao Paulo 01509-010, Brazil

**Keywords:** re-irradiation, salvage, recurrence, post-operative

## Abstract

*Background and Objectives*: Patients with recurrent squamous cell carcinoma of the head and neck (rHNC) face an aggressive disease. Surgical resection is the gold standard treatment. Immediate adjuvant post-operative stereotactic ablative radiotherapy (PO-SABR) for rHNC is debatable. *Materials and Methods:* We retrospectively identified patients who were treated with PO-SABR at the AC Camargo Cancer Center, Brazil. *Results:* Eleven patients were treated between 2018 and 2021. The median time between salvage surgery and PO-SABR was 31 days (range, 25–42) and the median PO-SABR total dose was 40 Gy (range, 30–48 Gy). The 2-and 4-year actuarial DFS were 62.3% and 41.6%, while the 2-and 4-year OS probabilities were 80.0% and 53.3%, respectively. Eight (72.7%) patients were alive and six (54.5%) were without disease at the last follow-up. Two (18.1%) patients had local failure in the PO-SABR field. Three (27.3%) patients had distant metastasis, diagnosed in a median time of 9 months (range, 4–13) after completion of PO-SABR. On univariate analysis, predictive factors related to worse OS were: interval between previous radiotherapy and PO-SABR ≤ 24 months (*p* = 0.033) and location of the salvage target in the oral cavity (*p* = 0.013). The total dose of PO-SABR given in more than three fractions was marginally statistically significant, favoring the OS (*p* = 0.051). *Conclusions:* Our results encourage the use of a more aggressive approach in selected patients with rHNC by combining salvage surgery with immediate PO-SABRT, but this association needs to be further explored.

## 1. Introduction 

Head and neck squamous cell carcinoma (HNC) is the sixth leading incident cancer worldwide and the 5-year overall survival (OS) remains less than 50%, despite advances in diagnosis and treatment, according to IARC’s GLOBOCAN database of national estimates [1,2]. 

Locoregional recurrence of HNC (rHNC) is a challenge and can occur in up to 65% of all patients treated [3]. Salvage therapy in this situation is a controversial issue and the best isolated approach or combination of treatment strategies is still to be defined, but recent advances in therapy have brought about substantial improvements in outcomes. For patients with a previous course of external beam radiation (EBRT) with infield recurrence, until now, radical surgical resection was considered the only curative salvage treatment, despite the fact that most patients refuse or are not suitable candidates for this approach [4]. Systemic therapy alone is associated with a median overall survival ranging from 5.0 to 10.1 months [5]. The combination of systemic therapies with re-irradiation for rHNC have also been published, but with poor outcomes [6]. Salvage re-irradiation with older techniques is also associated with low local control (LC), disease free (DFS), and overall survival (OS) rates. Two clinical trials, the RTOG 9610 and the RTOG 9911, which analyzed salvage re-irradiation plus chemotherapy, reported a median OS of 8.5 and 12.1 months, respectively [7,8]. 

New radiation technologies modalities such as stereotactic ablative radiation therapy (SABR), which is given with image-guidance, combining highly conformal and modulated irradiation techniques, allows for the administration of ablative doses to the tumor volume, minimizing the dose received by the normal surrounding tissue. The indication of local radiation with ablative doses, used as first line curative-intent therapy or as salvage treatment, is based on risk factors such as the number of metastases, interval to metastatic presentation, histology, and efficacy of systemic therapy [9], but the literature is still missing data regarding the association of immediate adjuvant post-operative SABR (PO-SABR) in rHNC. Our previous experience with favorable results using salvage interstitial high dose rate brachytherapy to improve LC and OS [6] has stimulated us to use PO-SABR as an adjuvant salvage treatment of rHNC in some situations, and therefore, the purpose of this review is to report on the clinical outcomes and major issues such as the toxicity profile and tolerability of patients who underwent PO-SABR as part of the salvage treatment at our Institution. 

## 2. Patients and Methods 

After approval of the institutional review board of the AC Camargo Cancer Center, Sao Paulo, Brazil, we retrospectively identified patients who were treated with adjuvant PO-SABR for rHNC. Inclusion criteria were: biopsy proven rHNC, absence of distant metastasis, history and interval between previous irradiation of at least 6 months, indication of curative salvage surgery intent, interval between surgery and adjuvant PO-SABRT up to 60 days, associated chemotherapy and or immunotherapy was allowed, and at least one clinical follow-up after the end of complete salvage treatment. Exclusion criteria were treatments with palliative intent and or total doses under 25 Gy or the use of dose fractions lower than 5 Gy. 

Salvage surgery consisted of *en bloc* tumor resection plus safety margins whenever possible. When the surgeon identified, at the moment of surgery, a possibility of marginal resection, metallic markers were inserted in the tumor bed to assist in defining it. No elective lymph node dissection was performed at the time of salvage surgery. Clinical characteristics including age, gender, risk factors (smoking and alcoholic intake), p16 status (for oropharynxes lesions) and performance status, associated co-morbidities, extent of disease, presence of metastatic disease, dose previously received, disease free interval from previous radiation, and associated systemic treatment were also evaluated. 

## 3. Radiation Plan and Dose Delivery 

After clinical evaluation and indication of PO-SABR, the patients underwent computed tomography (CT) plus magnetic resonance imaging (MRI) simulation with a custom thermoplastic head and neck mask for immobilization. Whenever possible, a PET-CT was performed and fused to the simulation CT images for target delineation.

The clinical target volume (CTV) was considered the tumor bed, as identified by the fusion of images at the moment of diagnosis of the local recurrence (CT, MRI, and or PET) with the planning TC plus expansion from 0 to 3 mm, depending on the anatomical site and or distance of the organs at risk, and did not include elective nodal coverage. In two (18.2%) patients, surgical resections left gross residual disease, one oral cavity and the other oropharynx (p16 negative), in these cases, the GTV was contoured and an expansion of 4 and 3 mm, respectively, was added to create the CTV. The planning target volume (PTV) was generated with a non-uniform expansion of 1 to 2 mm, depending on the anatomical location of the tumor bed. An example of a treatment plan for a left lymph node bed treated due to the presence of ECE is shown in Figure 1. 

Treatment planning was performed using Eclipse (Varian Medical Systems, Palo Alto, CA, USA). The plans were generated using the volumetric modulated arc therapy (VMAT) technique. Dose constraints were based on our previous experience using interstitial HDR [5], EQD2 dose calculations, and other references [10,11]. The critical organ constraints in our institutional protocol were (maximum dose summing the first and second course of radiation) for brachial plexus less than 105 Gy3. Different pontual maximum doses were accepted for the brain stem and spinal cord (Dmax 8 Gy), larynx, and mandible (Dmax 20 Gy) at PO-SABR. For other normal structures, the Institutional guidelines were followed. 

The daily setup consisted of cone-beam CT, previous to each treatment arch, using kV images. Treatments were delivered using a Varian TrueBeam Linac (Varian Medical Systems, Palo Alto, CA, USA). All treatments were scheduled to be given every other day over 2 weeks. 

## 4. Results 

There were 11 patients with a median age of 63 years old (range, 24–69), previously irradiated with infield recurrence. All underwent salvage surgery followed by PO-SABR between April 2018 and February 2021. The follow-up period ranged from 5.2 to 71.1 months (median—18.0 months). The median dose of previous EBRT given was 60 Gy (range, 40–70 Gy), in a range of 16 to 35 (median—30) fractions.

The median time between salvage surgery and PO-SABR was 31 days (range, 25–42) and the PO-SABR total dose ranged from 30 to 48 Gy (median, 40 Gy), given in two to six (median, 3) fractions.

Three (27.3%) patients presented with isolated nodal recurrences and had PO-SABR due to extra capsular extension. Excluding the nodal recurrences, the tumors were staged as rT1 (5–62.5%) and rT2 (3–37.5%). The demographic data of the patients and the key features of the initial treatment are depicted in Table 1. The pathologic findings at salvage surgery are described in Table 2. 

Five (45.4%) patients experienced disease progression after salvage therapy. Two (18.1%) patients had local failure in the PO-SABR field. Three (27.3%) patients had distant metastasis, diagnosed in a median time of 9 months (range, 4–13) after the completion of PO-SABR. 

The 2-and 4-year actuarial DFS were 62.3% and 41.6%, while the 2-and 4-year OS probabilities were 80.0% and 53.3%, respectively (Figure 2A,B). Eight (72.7%) patients were alive and six (54.5%) without evidence of disease at the last follow-up. 

The cumulative probability of local in field failure was 27.3% at the median follow-up time. The estimated median time to recurrence after PO-SABR was 15.3 months by the Kaplan–Meier product limit method (Figure 2B).

In the univariate analysis, predictive factors to worse OS were: interval between previous radiotherapy and PO-SABR ≤24 months (*p* = 0.033) and the location of the salvage target in the oral cavity (*p* = 0.013). The total dose of PO-SABR given in more than three fractions was marginally statistically significant with regard to the OS (*p* = 0.051). The univariate analysis data are shown in Table 3.

Physician-reported toxicity was according to the Common Terminology Criteria for Adverse Events (CTCAE) version 4.0. All patients developed at least Grade 1 acute mucosal toxicity. Seven (63.6%) patients developed acute Grade 2 local mucositis excluding the three who had lymph node bed irradiation and one who had a parotid bed irradiation. No patient developed severe acute or late toxicity up to this moment.

## 5. Statistical Analysis 

The survival analysis was performed using the Kaplan–Meier test and the Breslow statistic test was used to compare the differences in the survival curves. *p*-values of less than 0.05 were considered as statistically significant. SPSS v.25.1 (IBM, Armonk, NY, USA) for Windows was used for the statistical calculations. 

## 6. Discussion 

Several clinical characteristics and prognostic factors should be considered for patients undergoing salvage re-irradiation. The performance status as well as the location and extension of recurrent disease are the most important factors to be evaluated prior to an indication of a course of re-irradiation. It is imperative to respect the associated comorbidities that can also impact the survival and tolerability of salvage PO-SABR. Nonetheless, the interval from the previous irradiation to recurrence, the technique of radiation previously indicated as well as the doses received by critical structures and prior treatment toxicity must be carefully evaluated [5,10]. 

Local tumor growth can cause pain, infection, and bleeding aside from functional alterations of speech and swallowing. Cosmetic impact of the proposed treatment is also an important issue to be considered in planning a salvage treatment. Surgery is the main salvage strategy for rHNC, nevertheless, few patients are suitable candidates for curative resection and only 25% to 45% of patients experience long-term disease control [12]. For patients submitted to salvage surgery, the indications for a new course of post-operative radiation remain controversial, even in the setting of factors indicating that a high risk of local recurrence is present such as close or positive margins, extensive perineural invasion, worst pattern of invasion on the extension of surgical margins, lymph node extra capsular extension, or soft tissue infiltration [13].

Although many advances in diagnosis and treatment have been incorporated, metastatic disease, locorregional recurrences, or both, still develop in more than 65% of patients with HNC [3]. The concomitant incidence of distant metastasis for patients with rHNC is also high, but locorregional tumor progression is the predominant cause of death in these patients [14,15] and a major surgical approach cannot guarantee local control, especially when high risk factors for local recurrence, as cited above, are present.

Regarding adjuvant radiation treatment to salvage surgery, clinical data comparing the different radiation dose fractionation schemas are lacking in the setting of both radical and post-operative treatments with ablative radiation doses. 

Re-irradiation of infield failures via EBRT carries risks of increased toxicity. Using old techniques of radiation, it was accepted that doses in excess of EQD2 60 Gy were necessary to optimize the salvage probabilities in patients with rHNC, but the proximity of the recurrent lesion to critical structures has often limited the dose that could be safely given by re-irradiation and the reduced number of fractions. A wide range of SABR doses is described in the published literature and great differences exist when comparing the radiobiology, fractionation, and overall treatment time between EBRT and SABR. Vargo et al. compared the results of the re-irradiation of 217 patients treated with conventional fractionated IMRT (doses of at least 40 Gy) and 197 patients who had SABR (doses over 35 Gy, delivered in one to five fractions). In a subset analysis, they noted a similar OS, but the severe toxicity incidence was higher with IMRT (5.1% versus 0.5%, *p* < 0.01), suggesting a better tolerance of the previously irradiated tissue to higher doses per fraction [16]. 

When using ablative doses, the role of concurrent systemic therapy remains controversial. Some reports of salvage strategies combining systemic therapy and re-irradiation have shown improved LC, although at the expense of increased morbidity [7].

Nonetheless, even when using ablative and aggressive treatment schemas, a subset of patients can still experience no response to treatment. Two retrospective studies failed to show an improvement in LC with the addition of systemic therapy to SABR. Huang et al. explored the results of 74 patients treated from 2016 to 2019 at a single institute in Taiwan with 40–50 Gy in five fractions. SABR plus cetuximab was used in 16 (21.6%) patients. The 2-OS and DFS for all patients were only 22.0% and 19.0%, respectively. They observed that a re-irradiation interval >12 months was a statistical significance factor related to OS and LC [17]. Vargo et al. evaluated 132 patients treated between 2004 and 2011 with SABR given to a median dose of 44 Gy in five fractions. Seventy-two (55%) patients had concurrent cetuximab and SABR. They concluded that this combination had no impact on OS and LC. At a median 6-month follow-up (range, 0–55 months), they noted that a treatment duration less than 14 days was associated with improved LC and tumor volumes higher than 25 cc were linked to inferior OS and LC [18]. 

Diao et al. also explored the use of systemic therapy and or immunotherapy in association with SABR. They reviewed the charts of 137 patients who received SABR for rHNC from 2013 to 2020. The median SABR dose given was 45 Gy and the median target volume of 16.9 cc. Associated systemic therapy was used as per physician preference. They described that cetuximab dosing was 400 mg/m^2^ loading dose 1 week prior to radiation start, followed by 250 mg/m^2^ weekly during radiation. Adjuvant cetuximab was given weekly up to 15 cycles. Associated immunotherapy, when used, was with pembrolizumab (200 mg every 3 weeks) and or nivolumab (480 mg every 4 weeks). When using chemotherapy, concurrent cisplatin (60–80 mg/m^2^ every 3 weeks or 15–25 mg/m^2^ weekly) was given. The median OS was 44.3 months among all patients. The 2-year DFS was 32%. The 1-year local, regional, and distant controls were 78%, 66%, and 83%, respectively. They observed that systemic therapy improved the regional (*p* = 0.004) and distant control (*p* = 0.04) in non-metastatic patients at presentation [19]. Lartigau et al. published the results of 60 patients treated with SABR with doses of 36 Gy given in six fractions plus cetuximab. The 1-year OS rate was 47.5% [20]. Cengiz et al. followed 37 patients who were treated using the CyberKnife with a median of 30 Gy (range, 18–35 Gy) in one to five fractions. The ultimate LC was achieved in 31 (83.8%) patients, and the median OS was 12 months [21]. 

As described above, the association of systemic therapy and SABR is still controversial, and data regarding the association of salvage surgery and PO-SABR are missing in the literature. None of the patients in our series were treated with the combination of surgery, PO-SABR, and systemic therapy, which make it impossible to compare the results. 

Regarding toxicity, in our analysis, severe acute or late toxicity was not observed, probably because we carefully avoided including scars as targets and the use of boluses. 

The fractionation schemas for SABR published are very wide, which makes it very difficult to compare the data regarding the related toxicity. Some authors have already published the results and complications of SABR, but data on PO-SABR are scarce. Heron et al., in a phase I dose-escalation clinical trial, included 25 patients who were treated with 25 to 44 Gy given in five fractions every other day. They verified that 44 Gy given in five fractions was safe, with no Grade 3 or 4 toxicities observed [10]. Lartigau et al. reported one (1.7%) patient dead due to hemorrhage and malnutrition and 18 (30%) patients presenting with Grade 3 toxicities in a cohort of 60 patients treated with SABR (36 Gy given in 6 fractions) plus cetuximab [20]. Cengiz et al. observed in a group of 37 patients treated by CyberKnife (18–35 Gy given in one to five fractions) that eight (17.3%) patients had carotid blow-out syndrome, and seven (15.2%) patients died [21]. Hasney et al. published the results of 36 patients treated with doses ranging from 18 to 40 Gy (median, 30 Gy) and the incidence of complications were observed in three (8.3%) patients, one with bone necrosis and two with soft tissue necrosis [22]. 

Although patients with rHNC still face a relatively short OS, mainly due to uncontrolled local disease and metastatic progression, we could observe a higher median OS in our series than that reported by Siddiqui et al. [23]. Despite the higher number of patients in their series, when compared to our cohort (44 vs. 11), which they treated with either a single-fraction of 13–18 Gy or 36–48 Gy in five to eight fractions, the median OS they reported was 6.7 months when compared to 34.3 months in our series. 

Severe complications including death, mainly due to arterial morbidity, are not negligible in the setting of re-irradiation. Dose tolerance of the carotid to re-irradiation likely depends on the prior dose and on the time interval between courses. In this series, we took steps to minimize the dose to the carotid arteries, and did not observe vascular complications such as carotid blow-outs and aneurism formation. Grimm et al. reported that the most severe complications such as spinal cord myelopathy and carotid blowout, even the HyTEC data pooling efforts, were not currently sufficient to claim strong conclusions of the probability of risk [24].

Salvage surgery can result in close or positive surgical margins, increasing the risk of a second recurrence. In our series, three (27.3%) patients presented with close and two (18.2%) with positive margins, but no statistical significant differences in DFS and OS could be linked to these factors, probably due to the small number of patients evaluated. 

Similarly to our cohort, Hasney et al. presented their experience with the use of PO-SABR, but given by the CyberKnife. Six patients were treated due to close or positive margins. Five patients underwent re-irradiation of the cervical region and one of the peristomal neck. The average dose of radiation given was 23 Gy (Range 18–30 Gy) in five fractions. They observed in a follow-up of 8.5 months an OS rate of 83.3% (five/six patients) [22]. 

We also observed, in our analysis, a favorable statistically significant impact on the OS of an interval >24 months between the previous EBRT course and salvage PO-SABR (*p* = 0.033). Research evaluating the impact of the interval between the previous course and salvage re-irradiation has already being published. Kress et al. evaluated 85 patients treated between 2002 and 2011 with SABR with a median dose of 30 Gy. They noted that the median interval from the initial EBRT and PO-SABR was 32 months. The 2-year OS and DFS for patients treated with curative intent were 24% and 28%, respectively, compared to 80.0% and 62.3% in our series. As in our cohort, they also noted that an interval from the initial EBRT to SBRT of two years or more was associated with improved OS, *p* = 0.019 [25]. 

To the best of our knowledge, there have been few papers published on evaluating the results and indication of PO-SABR. Unfortunately, our small cohort prevented us from drawing any definitive conclusion regarding the impact of the anatomical site on OS. Even though the general outcome of our series was good and without significant severe acute or late toxicity associated, we suggest that further investigation is still necessary to define the best candidates for PO-SABR.

## 7. Conclusions 

Patients with rHNC face aggressive disease. Surgical resection is still the gold standard treatment in this situation. The selection of patients for a more intense regime of treatment will probably depend on the risk factors found in salvage surgery, which may positively impact the OS, DFS, and LC. Our results encourage the use of a more aggressive approach in selected patients by combining salvage surgery with PO-SABRT, but this association needs to be further explored.

## Figures and Tables

**Figure 1 medicina-58-01074-f001:**
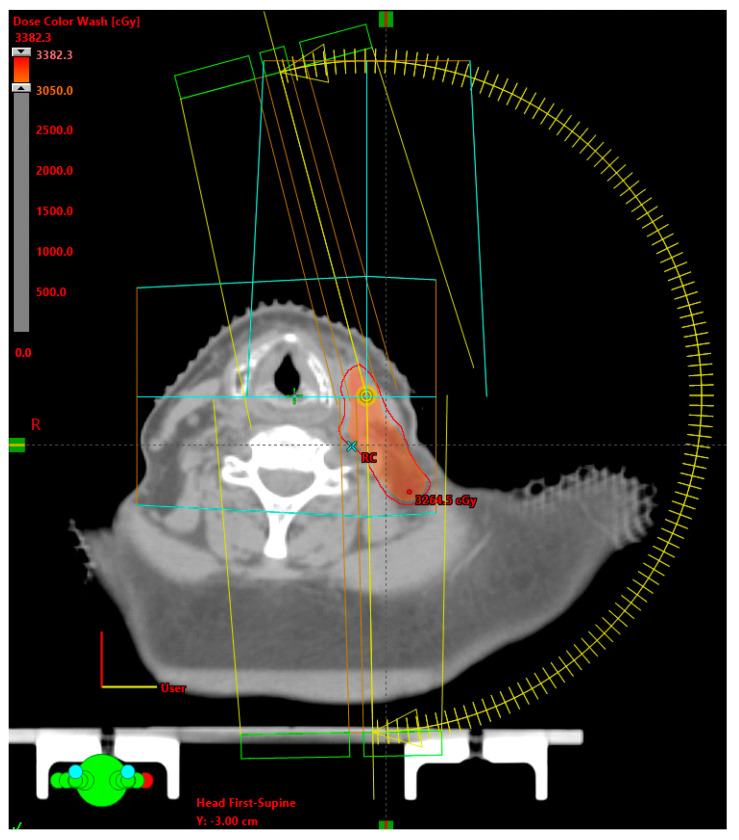
An example of a treatment plan for a left lymph node bed treated due to the presence of ECE.

**Figure 2 medicina-58-01074-f002:**
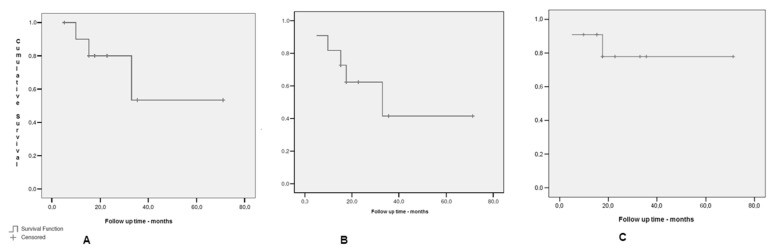
(**A**) Overall survival, (**B**) disease free survival and, (**C**) local control.

**Table 1 medicina-58-01074-t001:** The demographic data of the patients and the key features of the initial treatment.

Variable		Frequency (%)
Gender	Male	5 (45.5)
	Female	6 (54.5)
Anatomical location of infield of previous EBRT failure	Oropharynx SCC p16 positive	1 (9.1)
	Oropharynx SCC p16 negative	4 (36.4)
	Oral cavity SCC	2 (18.2)
	Salivary gland SCC	1 (9.1)
	Lymph node SCC metastasis	3 (27.3)
Dose at First EBRT course	≤60 Gy	4 (36.4)
	>60 Gy	7 (63.6)
Chemotherapy	Yes	8 (72.7)
	No	3 (27.3)
Interval between salvage surgery to the start of PO-SABR (days)	≤30	7 (63.6)
	>30	4 (36.4)
EBRT Technique	3D conformal	3 (27.3)
	IMRT	8 (72.7)

**Legend:** SCC–squamous cell carcinoma; EBRT–external beam radiotherapy.

**Table 2 medicina-58-01074-t002:** The pathological findings at salvage surgery.

Variable		Frequency (%)
Margin status	Free	6 (54.5%)
	Microscopic positive	3 (27.3%)
	Gross residual disease	2 (18.2%)
ECE	Yes	3 (27.3%)
	No	8 (72.7%)
Perineural Invasion	Yes	8 (72.7%)
	No	3 (27.3%)

**Legend:** ECE–extra capsular extension.

**Table 3 medicina-58-01074-t003:** The univariate analysis.

Variable		N	OS		DFS		LC	
			Censored	*p*	Censored	*p*	Censored	*p*
**Gender**	male	5	3	0.136	3	0.830	0	0.217
	female	6	0		2		2	
**Age (years)**	>60	7	1	0.546	3	0.663	1	0.763
	≤60	4	2		**2**		1	
**Anatomical** **Site**	Oropharynx SCC p16 negative	4	1	0.013	**2**	0.142	1	0.627
	Oropharynx SCC p16 positive	1	0		**0**		0	
	Oral cavity	2	2		**2**		0	
	Salivary gland	1	0		**0**		0	
	Lymph node	3	0		**1**		1	
**Margin status**	Free	6	2	0.529	2	0.459	0	0.186
	Microscopic positive	3	0		1		1	
	Gross residual disease	2	1		2		1	
**Perineural Invasion**		2			0.829	2	0.123	
	No	3	1		2		0	
**Previous EBRT total dose**	>60 Gy	7	2	0.658	3	0.666	2	0.251
	≤60 Gy	4	1		2		0	
**Interval between salvage surgery to the start of PO-SABR (days)**	≤30	7	2	0.248	3	0.722	1	0.117
	>30	4	1		2		1	
**PO-SABR total dose**	>40 Gy	9	1	0.078	3	0.222	1	0.102
	≤40 Gy	2	2		2		1	
**PO-SABR–number of fractions**	>3	8	1	0.051	3	0.341	0	0.465
	≤3	3	2		2		2	
**Interval first EBRT and salvage PO-SABR** **(months)**	>24	9	1	0.033	3	0.225	2	0.540
	≤24	2	2		2		0	
**Total**		11	3		**5**		2	

**Legend:** N—number of patients, OS—overall survival, DFS—disease free survival, LC—local control, SCC—squamous cell carcinoma, EBRT—external beam radiotherapy, PO-SABR—post-operative stereotactic ablative radiation therapy.

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
