# Peer review of "Salvage Post-Operative Stereotatic Ablative Radiotherapy for Re-Current Squamous Cell Carcinoma of Head and Neck"

_medicina, 2022, doi:10.3390/medicina58081074_

Round 1

Reviewer 1 Report

The work presents the single centre  experience on recovery with stereotaxic reirradiation for ablative purposes in case of recurrence of head and neck tumors. The work is a retrospective and descriptive effort. This issue is emerging thanks to the availability of increasingly precise techniques in defining the isodose in targeted fields. Another important aspect is the evaluation of toxicities which with these techniques can be minimized compared to more conventional methods.

The critical aspects of the work are attributable to the heterogeneity and limited case series and to  the different anatomical sites combined.  The positivity to p16 and / or high-risk HPV was not described (for the oropharynx), it is not clear whether the treatment was carried out after recovery surgery (as it would appear) or as conservative recovery only.

The results as reported in line 10 of abstract,  lines 13-14 and 102 are not clear ; sentence in line 212 it is not clear; it’s not clear  how many patients have undergone neck reirradiation. There is  there is a case of a salivary gland tumor), HPV.

References 7 and 18 are repeated in the bibliography.

The word preliminary results should be inserted in the title.

The work may be published with major revisions.

Author Response

defining the isodose in targeted fields.

Radiation Plan and Dose Delivery 

After clinical evaluation and indication of PO-SABR the patients underwent CT plus MRI simulation with custom thermoplastic head and neck mask for immobilization. Whenever possible, a PET-CT was performed and fused to the simulation CT images for target delineation.

The clinical target volume (CTV) was considered the tumor bed identified  by fusion of images at the moment of diagnosis of the local recurrence (TC, RM and or  PET) with the planning TC plus expansion from 0 to 3 mm, depending on the anatomical site and or distance of organs at risk, and did not include elective nodal coverage. In 2 (18.2%) patients surgical resections left gross residual disease, one in oral cavity and the other in oropharynx (p16 negative), in these cases the GTV was plotted and an expansion of 4 and 3 mm, respectively was added to create the CTV. The planning target volume (PTV) was generated with a non-uniform expansion of 1 to 2 mm, depending on the anatomical location of the tumor bed.

Another important aspect is the evaluation of toxicities which with these techniques can be minimized compared to more conventional methods.

Physician-reported toxicity was according to the Common Terminology Criteria for Adverse Events (CTCAE) version 4.0. All patients developed at least Grade 1 acute mucosal toxicity.  Seven (63.6%) patients developed acute grade 2 local mucositis, excluding those three who had lymph node bed irradiation and the one who had a parotid bed irradiation. No patient developed severe acute or late toxicity till this moment.

.  The positivity to p16 and / or high-risk HPV was not described (for the oropharynx),

recovery surgery (as it would appear) or as conservative recovery only.

Inclusion criteria were: biopsy proven rHNC, absence of distant metastasis, history and interval between previous irradiation of at least 6 months, indication of curative salvage surgery intent, interval between surgery and adjuvant PO-SABRT up to 60 days, associated chemotherapy and or immunotherapy was allowed and at least one clinical follow up after the end of complete salvage treatment.

Salvage surgery consisted of en bloc tumor resection plus safety margins whenever possible. When the surgeon identified at the moment of surgery a possibility of marginal resection, metallic markers were inserted in the tumor bed to assist in defining it. No elective lymph node dissection was performed at the time of salvage surgery.   Clinical characteristics including age, gender, risk factors (smoking and alcoholic intake), p16 status (for oropharynxes lesions) and  performance status, associated co-morbidities, extent of disease, presence of metastatic disease, dose previously  received, disease free interval from previous radiation, and associated systemic treatment were also evaluated.

The results as reported in line 10 of abstract,  lines 13-14 and 102 are not clear ; sentence in line 212 it is not clear; it’s not clear  how many patients have undergone neck reirradiation. There is  there is a case of a salivary gland tumor), HPV. corrected and clarified 

References 7 and 18 are repeated in the bibliography. corrected 

The word preliminary results should be inserted in the title. added 

The work may be published with major revisions.

Thanks for your suggestions 

Reviewer 2 Report

This is a study about post-operative stereotatic radiotherapy for recurrent squamous cell carcinoma of head and neck. The authors analyzed 11 patients with recurrent carcinoma.

In the Materials and methods section, the authors must described the statistical analyses. Did the authors use log-rank test to compare survival rates?

There are several grammatical errors. The authors should submit the text to a native English speaker to improve readability.

Please better describe how Gross Tumor Volume (GTV) was identified in the post-operative setting.

Some clinical parameters were not reported (e.g., TNM stage, HPV status, etc.).

Salvage surgery must be described.

PO-SABR total dose must be added.

Some images illustrating the irradiation fields may help the reader.

A more detailed description of adverse events should be added.

Author Response

A - Inserted Survival analysis was performed using Kaplan-Meier and  Breslow statistic test was used to compare differences in survival curves. P-values of less than 0.05 were considered statistically significant. SPSS v.25.1 for Windows was used for  statistical calculations. Breslow was used due to the small number in the cohort

B- There are several grammatical errors. The authors should submit the text to a native English speaker to improve readability. Done 

C - Please better describe how Gross Tumor Volume (GTV) was identified in the post-operative setting. Changed to: 

The clinical target volume (CTV) was considered the tumor bed identified  by fusion of images at the moment of diagnosis of the local recurrence (TC, RM and or  PET) with the planning TC plus expansion from 0 to 3 mm, depending on the anatomical site and or distance of organs at risk, and did not include elective nodal coverage. The planning target volume (PTV) was generated with a non-uniform expansion of 1 to 2 mm, depending on the anatomical location of the tumor bed.

D- Some clinical parameters were not reported (e.g., TNM stage, HPV status,

etc.).

Three (27.3%)  patients presented with isolated nodal  recurrences and had PO-SABR due to extra capsular extension. Excluding the nodal recurrences, the tumors were staged as  rT1 (5 - 62.5%) and rT2 (3 - 37.5%). HPV status was also inserted in tables 1 and 3

E - Salvage surgery must be described.

Salvage surgery consisted of en bloc tumor resection plus safety margins whenever possible. When the surgeon identified at the moment of surgery a possibility of marginal resection, metallic markers were inserted in the tumor bed to assist in defining it. No elective lymphonode dissection was performed at the time of salvage surgery.   

F - PO-SABR total dose must be added.

Median time between salvage surgery and PO-SABR was 31 days (range, 25-42) and PO-SABR total dose ranged from 30 to 48 Gy (median, 40 Gy), given in 2 to 6 (median, 3) fractions.

G A more detailed description of adverse events should be added.

Physician-reported toxicity was according to the Common Terminology Criteria for Adverse Events (CTCAE) version 4.0. All patients developed at least Grade 1 acute mucosal toxicity.  Seven (63.6%) patients developed acute grade 2 local mucositis, excluding those three who had lymph node bed irradiation and the one who had a parotid bed irradiation. No patient developed severe acute or late toxicity till this moment.

Reviewer 3 Report

The manuscript is a single institution review of head and neck cancer patients treated with SBRT.  Only 11 patients are described in the series, which is low but not unreasonable as all of the patients are re-irradiation cases that underwent surgical salvage prior to SBRT. SBRT is not typically done after surgical salvage, so the the topic is of some interest as such ablative treatment is usually based on treating tissue that is entirely gross disease or tissue that is not essential.  

The primary problem with the manuscript is that the authors do not describe how the target volumes are defined.  The section "Radiation Plan and Dose Delivery", line 67, should describe this. However, it is entirely based on targeting gross disease with a margin, and there is no gross disease in most of these cases (2/11 had residual macroscopic disease after surgery).  It is not at all clear how the surgical bed is targeted, and that is critical to evaluating the manuscript and interpreting the results.

The English needs a great deal of work.  There are numerous words, phrases, and expressions that appear to be inappropriate direct translations from another language.  The list below is by no means complete.

line 13: The 2- and 4-years....    This is an ambiguous sentence. Needs to be clarified. 

line 33: "facts" should be "fact"

Table 1: "lymphonode" should be "lymph node"

line 106: Estimated median time to recurrence estimated at 15.3 months based on Figure 1C (local control graph).  It appears this should be Figure 1B (DFS), because local control was about 80-90% at 15 months.

line 129: "deeply evalauted".  change "deeply" to "carefully".

line 139: "in special", change to "especially"

line 144: "Radiation therapy is seldom a component of initial treatment for HNC cancers, ..."   This is not true and makes no sense.

line 149: change "grate" to "great"

line 150: Sentence "Vargo....".    Unclear meaning.  "to" appears to be an unnecessary word, unclear what the authors are trying to state, and I have seen the paper they are referencing.

line 155:  potentialize is not a word.  "potentiate"?

line 164: "was a statistical significant factors..."  change factors to factor.

line 183: "motive of debate".  Change this to "controversial"

line 188: "It is important to note..."  sentence poorly written. Could be stated as "A wide range of SABR doses is described in the published literature", if this is what the authors are trying to state.

line 194: change "denutrition" to "malnutrition"

line 202: "Of notice is that 8...", change to "Notably, 8 (17.3%) patients had..."

line 214: "Despite the short f/u of 8.5 months they noted that 5/6 patients were alive (24)".  It is not surprising that survival is good with short followup, so the use of the word "despite" is inappropriate.

line 217: same problem as in the abstract, ambiguous language.

line 220: "The median OS they have reported was only 6.7 when compared to our results."   The authors to not compare to their results.

Author Response

The primary problem with the manuscript is that the authors do not describe how the target volumes are defined.  
Salvage surgery consisted of en bloc tumor resection plus safety margins whenever possible. When the surgeon identified at the moment of surgery a possibility of marginal resection, metallic markers were inserted in the tumor bed to assist in defining it. No elective lymphonode dissection was performed at the time of salvage surgery.   Clinical characteristics including age, gender, risk factors (smoking and alcoholic intake), p16 status (for oropharynxes lesions) and  performance status, associated co-morbidities, extent of disease, presence of metastatic disease, dose previously  received, disease free interval from previous radiation, and associated systemic treatment were also evaluated. 
Radiation Plan and Dose Delivery 
After clinical evaluation and indication of PO-SABR the patients underwent CT plus MRI simulation with custom thermoplastic head and neck mask for immobilization. Whenever possible, a PET-CT was performed and fused to the simulation CT images for target delineation.
The clinical target volume (CTV) was considered the tumor bed identified  by fusion of images at the moment of diagnosis of the local recurrence (TC, RM and or  PET) with the planning TC plus expansion from 0 to 3 mm, depending on the anatomical site and or distance of organs at risk, and did not include elective nodal coverage. In 2 (18.2%) patients surgical resections left gross residual disease, one in oral cavity and the other in oropharynx (p16 negative), in these cases the GTV was plotted and an expansion of 4 and 3 mm, respectively was added to create the CTV. The planning target volume (PTV) was generated with a non-uniform expansion of 1 to 2 mm, depending on the anatomical location of the tumor bed.

The English needs a great deal of work.  There are numerous words, phrases, and expressions that appear to be inappropriate direct translations from another language.  The list below is by no means complete.
Reviewed

line 13: The 2- and 4-years....    This is an ambiguous sentence. Needs to be clarified. 
The actuarial projected 2- and 4- years DFS and OS probabilities are 62.3%, 41.6%, 80.0% and 53.3%, respectively. 
l
ine 33: "facts" should be "fact" --> corrected

Table 1: "lymphonode" should be "lymph node" --> corrected

line 106: Estimated median time to recurrence estimated at 15.3 months based on Figure 1C (local control graph).  It appears this should be Figure 1B (DFS), because local control was about 80-90% at 15 months.
corretcted
line 129: "deeply evalauted".  change "deeply" to "carefully".
done
line 139: "in special", change to "especially"

line 144: "Radiation therapy is seldom a component of initial treatment for HNC cancers, ..."   This is not true and makes no sense.
Re-irradiation of infield failures via EBRT carries risks of increased toxicity. (do you agree with this prase? 
line 149: change "grate" to "great" - done 

line 150: Sentence "Vargo....".    Unclear meaning.  "to" appears to be an unnecessary word, unclear what the authors are trying to state, and I have seen the paper they are referencing.
Corrected
line 155:  potentialize is not a word.  "potentiate"? yes, corrected

line 164: "was a statistical significant factors..."  change factors to factor.corrected

line 183: "motive of debate".  Change this to "controversial" done

line 188: "It is important to note..."  sentence poorly written. Could be stated as "A wide range of SABR doses is described in the published literature", if this is what the authors are trying to state.
Changed to:  A wide range of SABR doses is described in the published literature and great differences exists when comparing the radiobiology, fractionation and overall treatment time between EBRT and SABR.

line 194: change "denutrition" to "malnutrition" done

line 202: "Of notice is that 8...", change to "Notably, 8 (17.3%) patients had..."

line 214: "Despite the short f/u of 8.5 months they noted that 5/6 patients were alive (24)".  It is not surprising that survival is good with short followup, so the use of the word "despite" is inappropriate.
They observed in a follow-up of 8.5 months an OS rate of 83.3% (5/6 patients

line 217: same problem as in the abstract, ambiguous language.
On univariate analysis, predictive factors related to worse OS were: interval between previous radiotherapy and PO-SABR < 24 months (p=0.033) and location of salvage target in the oral cavity (p=0.013). Total dose of PO-SABR given in more than 3 fractions was marginally statistical significant favoring the OS (p=0.051).

We have also observed in our analysis a favorable statistical significant impact on OS of an interval >24 months between the previous EBRT course and salvage PO-SABR (p=0.033). Research evaluating the impact of the interval between previous course and salvage re-irradiation has already being published. Kress et al. evaluated 85 patients treated between 2002 and 2011 with SABR with a median dose of 30 Gy. They noted that the median interval from initial EBRT and PO-SABR was 32 months. The 2-year OS and DFS for patients treated with curative intent were 24% and 28%, respectively, compared to 62.3% and 80.0% in our series. As in our cohort, they also noted that an interval from initial EBRT to SBRT of 2 years or more was associated with improved OS, p = 0.019 (25). 

line 220: "The median OS they have reported was only 6.7 when compared to our results."   The authors to not compare to their results. corrected

I want to thank you for helping to improve the language. 

Round 2

Reviewer 1 Report

The revised work corresponds to an exact description of the sample studied, the subdivision of the anatomical specifications and characteristics of the neoplasia are consistent with the treatment carried out and with the results obtained. Several biases remain due to the heterogeneity of the sample and the timing of the treatment. The work is however of scientific interest in light of the increasing use of radio-targeted recovery technique

Author Response

English language and style are fine/minor spell check required --> all minor revisions pointed out have been fixed

Reviewer 2 Report

Thank you for improving the manuscript.

Author Response

English language and style are fine/minor spell check required--> all minor revisions pointed out have been fixed

Reviewer 3 Report

The authors have made extensive corrections to the language and content.  Some problems remain.

1) line 19: "The 2- and 4-years actuarial DFS and OS probabilities were 62.3%, 19 41.6%, 80.0% and 53.3%, respectively."   This remains ambiguous, pointed out earlier, and it does not appear that the authors understand why.  Are the first 2 numbers DFS at 2 and 4 years, or the 2 year DFS and OS?

2) Problem 1 above is repeated in the text, line 145.

3) line 38.  Change "controversy" to controversial.

4) line 41 (and throughout).  Change "till" to until.

5) line 62.  Change "indicate" to "use" or "utilize"

6) line 93 & 94: abbreviations "TC" and "RM" are used, and are not defined earlier.  From context I would gues these mean "CT" and "MRI", but not clearly stated.

7) line 96 & 97: removed the word "in" before oral cavity and before oropharynx.

8) line 97, change "plotted" to "contoured"

9) line 108.  Change "the critical organs constraints" to "the critical organ constraints"

10) line 111.  The 2 Gy equivalent Dmax permitted for the mandible is quoted at 20Gy, and it is stated that this is the summation of first and second courses of RT. This is not credible.  Similarly for the 8Gy Dmax for the spinal cord.  Both doses were likely exceeded by the first course alone.

11) line 159.  Change "favoring the OS" to "with regard to OS"

12) line 178.  Sentence beginning "It is impossible...".  Change this to "It is imperative to respect the associated comorbidities that can also impact ...."

13) line 193. "it is still controversial...."   Change this to "the indications for a new course of post-operative radiation remain controversial, even in the setting of factors indicating a high risk of local recurrence are present, such as....."

14) line 199.   Change "develops" to "develop"

15) line 213.  change "exists" back to "exist"

16) line 220: change "in the IMRT" to "with IMRT"

17) line 222: Paragraph poorly written.  Change to: When using ablative doses, the role of concurrent systemic therapy remains controversial.  Some reports of salvage strategies combining systemic therapy and re-irradiation have shown improved LC, although at the expense of increased morbidity (7).

18) line 236: change "association of" to "concurrent"

19) line 237: change "this association" to "combination therapy"

20) line 264.  English.  Sentence starting with "There was no patient..."   Change to "None of the patients in our series were treated with the combination of surgery, PO-SABR, and systemic therapy."

21) line 294:  change "when talking about" to "in the setting of"

22) line 294: The authors state that they took great care to avoid re-irradiation close to the carotid.  Yet the example in figure 1 partially includes the carotid in the target volume.  This sentence could be stated less strongly: "In this series we took steps to minimize dose to the carotid arteries, and did not observe vascular complications such as carotid blow-outs and aneurism formation".  

23) same paragraph as in #22 above: the HyTEC article on carotid blowout after SBRT should be referenced (Grimm et al, Red Journal, 2021, "Initial Data Pooling....")

24) lines 328 and 329.  OS and DFS numbers are quoted, with DFS higher than OS.  This does not make sense, and in the case of the data from this manuscript, does not agree with the graphs present earlier.

25) line 335. Replace end of sentence "... severe acute and ...' with "... severe acute or late toxicity, further research is necessary to define appropriate candidates for PO-SABR.

Author Response

1) line 19: "The 2- and 4-years actuarial DFS and OS probabilities were 62.3%,  41.6%, 80.0% and 53.3%, respectively."   This remains ambiguous, pointed out earlier, and it does not appear that the authors understand why.  Are the first 2 numbers DFS at 2 and 4 years, or the 2 year DFS and OS?

2) Problem 1 above is repeated in the text, line 145.

The 2-and 4-years actuarial DFS were 62.3%  and 41.6%,  while the 2-and 4-years OS probabilities were 80.0% and 53.3%, respectively. 

3) line 38.  Change "controversy" to controversial.--> copied lines 36-39. The word is not in the phrases:

 a previous course of external beam radiation (EBRT) with infield recurrence, untill now, the surgical radical resection was considered the only curative salvage treatment, despite the fact that most patients refuse or are not suitable candidates for this approach (4). Systemic therapy alone is associated with a median overall survival ranging

4) line 41 (and throughout).  Change "till" to until.--> done  

5) line 62.  Change "indicate" to "use" or "utilize" done

 line 93 & 94: abbreviations "TC" and "RM" are used, and are not defined earlier.  From context I would gues these mean "CT" and "MRI", but not clearly stated. Corrected --> and inserted a description on first presentation 

7) line 96 & 97: removed the word "in" before oral cavity and before oropharynx. done

8) line 97, change "plotted" to "contoured" done

9) line 108.  Change "the critical organs constraints" to "the critical organ constraints" done

10) line 111.  The 2 Gy equivalent Dmax permitted for the mandible is quoted at 20Gy, and it is stated that this is the summation of first and second courses of RT. This is not credible.  Similarly for the 8Gy Dmax for the spinal cord.  Both doses were likely exceeded by the first course alone.

. Different pontual maximum doses were accepted for the brain stem and spinal cord (Dmax 8 Gy), larynx and  mandible (Dmax 20Gy) at PO-SABR . 

11) line 159.  Change "favoring the OS" to "with regard to OS" done

12) line 178.  Sentence beginning "It is impossible...".  Change this to "It is imperative to respect the associated comorbidities that can also impact ...."

Done . It is imperative to respect the associated comorbidities that can also impact survival and tolerability of salvage PO-SABR. Nonetheless

13) line 193. "it is still controversial...."   Change this to "the indications for a new course of post-operative radiation remain controversial, even in the setting of factors indicating a high risk of local recurrence are present, such as....."

For patients submitted to salvage surgery the indications for a new course of post-operative radiation remain controversial, even in the setting of factors indicating a high risk of local recurrence are present,  such as close or positive margins, extensive perineural invasion, worst pattern 

14) line 199.   Change "develops" to "develop" done 

15) line 213.  change "exists" back to "exist" done 

16) line 220: change "in the IMRT" to "with IMRT" done

17) line 222: Paragraph poorly written.  Change to: When using ablative doses, the role of concurrent systemic therapy remains controversial.  Some reports of salvage strategies combining systemic therapy and re-irradiation have shown improved LC, although at the expense of increased morbidity (7) done 

18) line 236: change "association of" to "concurrent" - done

19) line 237: change "this association" to "combination therapy" - done

20) line 264.  English.  Sentence starting with "There was no patient..."   Change to "None of the patients in our series were treated with the combination of surgery, PO-SABR, and systemic therapy."

21) line 294:  change "when talking about" to "in the setting of"- Severe complications, including death, mainly due to arterial morbidity, are not negligible in the setting of re-irradiation. 

22) line 294: The authors state that they took great care to avoid re-irradiation close to the carotid.  Yet the example in figure 1 partially includes the carotid in the target volume.  This sentence could be stated less strongly: "In this series we took steps to minimize dose to the carotid arteries, and did not observe vascular complications such as carotid blow-outs and aneurism formation".

23) same paragraph as in #22 above: the HyTEC article on carotid blowout after SBRT should be referenced (Grimm et al, Red Journal, 2021, "Initial Data Pooling....")

Severe complications, including death, mainly due to arterial morbidity, are not negligible in the setting of re-irradiation. Dose tolerance of the carotid to re-irradiation likely depends on the prior  dose and on the time interval between courses.  In this series we took steps to minimize dose to the carotid arteries, and did not observe vascular complications such as carotid blow-outs and aneurism formation. Grimm et al. reported that the most severe complications such as spinal cord myelopathy and carotid blowout, even the HyTEC data pooling efforts are not currently sufficient to claim strong conclusions of probability of risk.

Grimm J, Vargo JA, Mavroidis P, Moiseenko V, Emami B, Jain S, Caudell JJ, Clump DA, Ling DC, Das S, Moros EG, Vinogradskiy Y, Xue J, Heron DE. Initial Data Pooling for Radiation Dose-Volume Tolerance for Carotid Artery Blowout and Other Bleeding Events in Hypofractionated Head and Neck Retreatments. Int J Radiat Oncol Biol Phys. 2021 May 1;110(1):147-159.

24) lines 328 and 329.  OS and DFS numbers are quoted, with DFS higher than OS.  This does not make sense, and in the case of the data from this manuscript, does not agree with the graphs present earlier. corrected

25) line 335. Replace end of sentence "... severe acute and ...' with "... severe acute or late toxicity, further research is necessary to define appropriate candidates for PO-SABR.

Thanks again for suggestions that improved the understanding of the paper !
